# Designing Socially Assistive Robots for Alzheimer’s Disease and Related Dementia Patients and Their Caregivers: Where We Are and Where We Are Headed

**DOI:** 10.3390/healthcare8020073

**Published:** 2020-03-26

**Authors:** Dimitrios Koutentakis, Alexander Pilozzi, Xudong Huang

**Affiliations:** Department of Psychiatry, Massachusetts General Hospital and Harvard Medical School, Charlestown, MA 02129, USA; dkout@mit.edu (D.K.); apilozzi@mgh.harvard.edu (A.P.)

**Keywords:** aging, dementia, alzheimer’s disease, socially assistive robots

## Abstract

Over the past few years there has been a large rise in the field of robotics. Robots are being in used in many industries, but there has not been a large surge of robots in the medical field, especially the robots for healthcare use. However, as the aging population keeps growing, current medical staff and healthcare providers are increasingly burdened by caring for the ever-growing number of senior patients, especially those with cognitive impairment of Alzheimer’s disease (AD) and Alzheimer’s disease-related dementia (ADRD) patients. As a result, we can expect to see a large increase in the field of medical robotics, especially in forms of socially assistive robots (SARs) for senior patients and healthcare providers. In fact, SARs can alleviate AD and ADRD patients and their caregivers’ unmet medical needs. Herein, we propose a design outline for such a SAR, based on a review of the current literature. We believe the next generation of SARs will enhance health and well-being, reduce illness and disability, and improve quality of life for AD and ADRD patients and their caregivers.

## 1. Introduction

The field of robotics has broad applications throughout various industries. In the medical sector, surgical robots are quite common in hospitals, starting with the early da Vinci and Zeus models [1]. Despite this, the use of service robotics in the medical field lags behind other industry sectors, accounting for only 3% of total robot sales [2,3]. A summary figure of the use of service robots per industry can be found in Figure 1.

Though there are research and developmental efforts geared towards robotics applications in diagnostics and certain types of physical therapy [5,6], many areas of the medical sector that may benefit from automation have not seen successful development of robotic solutions. As an aging population increasingly demands more care from a diminishing pool of caregivers, automated systems have been proposed, but not widely implemented, for taking over some of the social and assistive services for caregivers provide. Based on patient and caretaker feedback, we propose a primarily user-centered methodology for designing a socially assistive robot (SAR). Herein, we describe the increasing challenges posed by Alzheimer’s disease and related dementias, the implementation and testing of SARs and other assistive robots and what benefits they can provide, and how the acceptance of a household or care-home robot can be maximized. References were obtained primarily by searching for key terms on PubMed.gov (e.g., “dementia”, “robot”) and filtering by title and abstract for non-case studies relevant to dementia or other forms of cognitive impairment. References were also harvested from review papers and other manuscripts.

### Alzheimer’s Disease and Dementia

Alzheimer’s disease is one of the most prevalent neurodegenerative diseases in older adults, and is the most common cause of dementia. It is estimated that 10% of persons over the age of 65 have Alzheimer’s dementia, and this percentage increases with age; roughly 32% of those over the age of 85 suffer from the disease [7]. The cognitive symptoms of dementia in general are varied, and notably include memory loss and learning difficulties, among other psychiatric, mood, and behavioral disturbances [8]. Therapeutic techniques for treating AD are mostly limited to targeting specific symptoms of the disease, without addressing the root causes of the neurodegenerative process [9]. This is further exacerbated by the difficulty in diagnosing the disorder, as confirmation can only truly be found in postmortem analysis of the patient’s brain; reliable biomarkers are a subject of considerable research imported for this reason [10].

As the global population ages, it is expected that there will be a corresponding increase in patients with symptoms of dementia. The number of people affected by dementia in 2018 was around 47 million, and it is expected that this number will increase to over 75 million by 2030 [11]. The large quantity of dementia patients and the scale of care required to treat them has created a significant socioeconomic burden. The estimated cost incurred by dementia in 2015 was approximately US $957 billion, and this cost is projected to balloon to US $2.54 trillion in 2030, and US $9.12 trillion by 2050 [12]. It therefore becomes evident how important it is to find solutions that will facilitate the care of the rapidly increasing population of older people with dementia. Indeed, increases in life expectancy and decreases in the birth rates of many countries, as shown in Figure 2, mean that the increase in the number of older individuals outpaces the overall population growth rate [13,14]. As a result of this, the ratio of persons aged 20–64 to those over the age of 64 decreases, and so too will the ratio of caregivers to patients [15]. Since AD is the most common form of senile dementia, it is imperative to aim for any such solution to Alzheimer’s Disease and Related Dementia (ADRD) patients and their caregivers. Socially assistive robotics (SARs) are a promising avenue to provide care for the large numbers of ADRD patients and alleviate some of the burden placed on their caregivers.

## 2. Assistive Robotics

### 2.1. Existing Solutions

The subcategory of robots that serve to provide social facilitation and companionship to users are Socially Assistive Robots (SARs). SARs are the intersection between Assistive Robots and Socially Interactive Robots, with the goal to provide assistance to human users through social interaction [18]. Attempts to develop SARS for interaction with dementia patients are not uncommon. So-called social commitment robots have been developed, with some taking the form of relatively simple pseudo-animal robots made of metal or with synthetic fur, such as NeCoRo, AIBO, and Paro, which represent a cat, dog, and baby harp seal, respectively, have been implemented in elderly care-homes [19]. Other more complex implementations of the SAR concept include “Bandit”, an anthropomorphic robot representing the human torso and upper body featuring more advanced social queues, such as pointing, some speech, and interaction via a simple game [20].

Another area of considerable interest for applying robotics to dementia care comes in the form of cognitive assistive robots and technologies that aid users in performing activities of daily living. Such devices typically have roles in helping patients to maintain a schedule and remember certain tasks. More complicated models use computer-vision to determine what the user is trying to do, and prompt them to perform the task correctly. One example is the “COACH” system that was designed to assist dementia patients with handwashing. The system is capable of recognizing handwashing steps, as seen through a camera, and prompts the user with audio and shows video demonstrations as necessary. The system increased the ability of most subjects to complete handwashing steps independently, and reduced the need for caregiver intervention [21]. A robot prototyped in 2013 was developed to similarly identify steps, and intervene as needed, in the process of making tea. Though most of the participants in this study had the cognitive capacity to complete the task themselves, most subjects and their caregivers reported interest in the robot, noting that it was useful, and would be nice to have at a later stage of the subject’s disease [20]. Other implementations of this paradigm include Nursebot and the Autominder system [22].

### 2.2. Efficacy of SARs

Though research in the field is somewhat sparse, there is a body of work that suggests SARs are effective in improving the lives of those living with dementia. A 2004 study that introduced NeCoRo, a robotic cat featuring synthetic fur, lifelike behaviors, and an ability to respond to user touch [23], to 19 (10 male, 9 female) residents of a nursing home for dementia patients observed a decrease in general agitation. The effect was found to be comparable to that of a simple plush-toy cat, however [24]. An early iteration of Artificial Intelligence Robot (AIBO), a non-furry robotic dog able to respond to touch and voice commands, was introduced to 13 patients (1 male, 12 female) with dementia staying at a geriatric care facility in 2004. Interest in AIBO increased over time, and the presence of the robot was observed to increase social interactivity between the other subjects in the study, demonstrating a social-faciliatory effect. However, much like NeCoRo, the observed effects were comparable to a furry toy dog exhibiting simple puppy-like behaviors; the toy dog also required less prompting by the observing occupational therapist for the subjects to interact with the item. Development of an attachment to AIBO was also limited [25]. A second study introducing AIBO to patients of a long-term care home, who do not necessarily have any degree of dementia, found that subjects were able to develop attachments to the robot, and exhibited significant decreases in loneliness; these benefits were lesser than those of a real dog, however. Interestingly, the decrease in loneliness was not observed to correlate to the level of attachment, indicating that the development of attachment is not necessary to see other benefits [26].

One of the more successful implementations of an animal-like SAR is Paro, the robotic baby harp seal. Paro is equipped with synthetic fur, and a variety of sensors that allow its behavior to change based on user input. It has the ability to modify its facial expressions, and direct its “focus” towards stimuli such as noise. It is notably capable of long-term reinforcement-based learning, with actions such as petting to reinforce a behavior, while hitting the robot will discourage a behavior; the robot can also learn a new name for itself if it hears it frequently [27]. Introducing the robot into a care home with primarily high-functioning elderly individuals showed mixed results. Some responded well to Paro, while others developed limited attachments. Overall social interaction between study subjects increased in the presence of Paro and a caregiver. Notably, this study made Paro available to small groups every two weeks for a period of 4 months, making its presence somewhat sporadic for the residents [28]. A 2013 study comparing the effects of Paro to those of a resident dog presented Paro to care-home residents twice a week over the course of 12 weeks. The researchers reported that the robot significantly decreased loneliness in residents, was interacted with, and stimulated conversation between residents more than the resident dog [29]. Peterson et al. utilized Paro in 20-minute sessions three times per week with mild–moderate dementia patients. Lessened stress, depressive symptoms, and pain in those treated with Paro was noted compared to standard nurse-led activities [30]. Moyle et al. reported that Paro improved neutral affect, apparent pleasure, and observed agitation when studying dementia patients. However, the effect on neutral affect was comparable to a plush toy, and there was no significant difference in agitation when measured using the Cohen–Mansfield Agitation Inventory Short Form [31]. This study was also not the only one to find Paro to be in some ways comparable to that of a stuffed toy; Mervin et al. looked at agitation in dementia patients following the use of Paro or a plush toy, and found no significant difference between the two treatment groups [32]. Overall, it seems that animal-like SARs can be beneficial in certain circumstances, but their efficacy and cost-effectiveness compared to alternatives is questionable.

Looking at more complex social robots specifically, some relatively recent studies make use of the Nao robot, which consists of a small, humanoid chassis with an easily modified programming architecture. One study involved a Nao model programmed to engage patients in various types of therapy, namely language, music, and physiotherapies, as well as storytelling sessions. Results showed promise, with a high degree of acceptance among patients, and the robot was able to consistently capture patient attention with its humanoid appearance, movement, and other capabilities. Simpler interactions were programmed for use with severe dementia patients, as they did not maintain attention well in the structured sessions, highlighting the necessity for different development strategies depending on the severity of dementia [33]. A pilot study for a novel, customizable dementia-companion robot named the “Ryan Companion-bot” was given to subjects in their private rooms 24/7 for a period of 4–6 weeks. The robot featured a head-projection system that allowed for a human-like face with facial expressions and speech visualization, along with a touch screen display for certain interactions and cognitive games. It was also capable of speech and facial-expression recognition, and could hold conversations with users about certain topics. Subjects consistently spent an average of over 2 hours per day with Ryan, frequently conversing with the robot and taking advantage of its various interactive features; most users expressed interest in keeping Ryan for an extended period of time [34]. Chu et al. introduced two uniquely shaped social robots, named “Sophie” and “Jack”, to dementia care home residents. The robots featured the ability to track subjects, and interact through gestures and vocalizations; the robot could understand speech, but it did not have its own conversational ability. Aside from individual interaction, the robots also engaged in singing, and other activities with patients. The authors reported that the subjects responded positively to the two robots, and patient engagement with the robots improved significantly over the course of the multi-year study [35]. Other models will be discussed in the following sections.

Overall, it seems that there are sufficient benefits from robot-assisted activities to warrant further investigation. However, it is clear that the benefits these robots can provide is determined by the individual characteristics of the patient. Liang et al. reported that the response of dementia patients to Paro was dependent on the degree of cognitive impairment, with more cognitively impaired individuals responding more poorly to the robot than those experiencing more mild impairment [36]. Specifically looking at those with moderate-severe dementia, a study involving therapy dogs, Paro, and a humanoid robot called Nao in the treatment of patients found that the robots were less effective than conventional therapies. Furthermore, Paro in particular appeared to worsen certain behaviors and decrease quality of life [37]. On the other hand, Wada et al. found that those individuals nearing cognitive normalcy benefited from robots like Paro, and the nurses responsible for caring for the patients reported lower burnout than those without the assistance of Paro [38]. However, somewhat contrastingly, a study published in 2020 found that older individuals with more severe dementia experienced greater benefits from communication robots than did younger individuals with mild dementia [39]. Joranson et al. also reported benefits from Paro in those with severe dementia, indicating that the degree of cognitive impairment alone dictates the efficacy of robot therapy [40]. Lane et al. notes that Paro is effective in improving mood and behavior in calm, approachable individuals, while those who are actively experiencing behavior or mood problems respond poorly to the robot [41]. It would seem that both a patient’s cognitive status and personality will need to be considered when utilizing robots in dementia treatment.

### 2.3. Acceptance, Needs, and Preferences

In order for SARs and other assistive robot technologies to find use in the care of dementia patients, they will need to be accepted by the patients themselves. As an example, researchers of the COACH system noted that one particular participant, who was the most independent of the research subjects, completely ignored COACH, and only accepted instruction from their caregiver [21]. Thus, it is imperative that efforts be made to tailor any robotic-care solution to the perspectives of the users, namely, elderly individuals with some degree of cognitive impairment.

Much insight into the desires and preferences of potential users and adopters of a SAR comes from various surveys administered to elderly individuals with some cognitive impairment, as well as their caregivers and medical personnel. Much consideration is given to what functions patients and those who care for/treat them believe they should have. A 2017 survey conducted on potential users, 20% of whom had early-stage AD and the rest had some degree of mild cognitive impairment (MCI), caregivers, and medical staff found varying priorities between groups. The functions considered the highest priority by all groups were related to patient safety, such as features that call for help if something happens to the patient, detects obstacles for fall prevention, ensure that water, gas, and lights are turned off, as well as features that ensure patients take their medications properly [42]. Similar safety and healthcare features were rated as high-priority by a smaller survey conducted in Paris, published in 2015, of 25 elderly individuals, though not as highly as some other functions. It should be noted that some participants in this survey were cognitively healthy and/or were informally caring for someone with cognitive impairment [43]. A 2010 survey of dementia caregivers on smart-home technology found that the most popular features were related to safety and monitoring, highlighting the need for a system that can protect elderly dementia patients, and alleviate some of the worry of caregivers [44].

The Paris survey found that functions that provide cognitive support and communication were among the most desired functions of a robotic assistant [43]. Though such functions were not deemed to be as important as safety-related features in the Spain–Poland survey, assistance in finding objects, stimulation of interaction with family/friends, and the provision of cognitive exercises were rated as being high-priority, though potential users rated assistance with their daily tasks as generally low-priority. Interestingly, caregivers and medical personnel rated many of these tasks as medium–high priority, indicating a discrepancy between what potential users think they need, and what those who care for them believe would be useful; potential users may not want to admit that assistance with certain functions would be beneficial. In a similar vein, fewer potential users believed that the robot should be able to identify the user’s emotional state and act to try to improve that state than did caretakers and medical staff [42]. Furthermore, while it was agreed that some of these functions should be autonomously engaged by the robot, such as safety-relevant functions, other features, such as cognitive stimulation, were desired to be on-demand by potential users, while caretakers/medical staff believed the robot should initiate the behavior [42]. Most of the issues deemed high-priority by patients and/or caregivers/staff can be readily solved using computer vision, and simpler technologies, such as automatic reminders and timers.

The interface of the robot is a subject of considerable importance. The “Guide” robot, which is a stationary robot 1.6 meters tall with a touchscreen interface featuring utilitarian applications, as well as entertainment and cognitive games, was criticized by users and relatives of dementia patients for being too complicated to use. Some functions, such as inputting the user’s name, were too difficult for patients to handle; the height and slant of the screen also made it difficult for patients to manipulate [3]. More complicated social robots that use voice recognition and, in some cases, facial expressions and body language to communicate are generally well-received by subjects and caregivers [35,45]. In the aforementioned Spain–Poland survey, users, caregivers, and medical staff greatly preferred voice-based communication to other options, and voice communication is the most natural and simple way of interaction for most patients. Notably, touch screens were the next preferred interface for users, though roughly half as popular as voice commands. Medical staff in particular believed that the ability to issue commands via simple gestures was a high priority, though this belief was not shared by the caregivers, nor the patients themselves [42]. It is probable that gesture-based communication would be most useful for the more severely impaired, something that is perhaps more recognized by medical professionals.

The appearance of the robot also needs to be tailored to users’ expectations. The overwhelming majority of potential users and caregivers alike in the Spain–Poland survey believed that the assistive robot should be as tall as, or shorter than the user, with a preference towards chest height [42]. The Paris survey found that mechanical-looking human-like robots were the most popular, with those which were mechanical animal-like, animal-like, and machine-like receiving roughly 20% popularity each [43]. Android/human-like robots were decidedly unpopular, perhaps due to the “uncanny valley” effect evoked by entities that are close to, but not exactly, human in appearance [46,47]. The majority of those surveyed in the Spain-Poland survey similarly believed an anthropomorphic robot would be best, though no preference in material makeup of the robot was noted [42].

With regards to overall robot acceptance in the Spain-Poland surveyed group, more than 80% of the participants believe that “it is a good idea to replace a human caregiver with a robot”, and more than 75% of caregivers would agree to leave a user alone with a robot [42]. When the potential-user group was asked if they would like an SAR of their own, most participants stated that they would be more willing to use SARs in the future rather than in the present, signifying some skepticism in their utility, while caregivers were more concerned with usability issues. However, both caregivers and MCI patients agreed that an SAR could provide useful cognitive support [43]. Introduction of patients to robot assistants, however, shows some promising results. In a small study published in 2017, the acceptance of assistive robots was tested by having patients interact with a tele-operated robot. The study evaluated the reactions of the 10 mild-to-moderate AD patients and their 10 respective caregivers. While both the caregivers and the patients seemed interested in the technology to an extent, patient interest was lacking, mainly because the patients showed a lack of perceived need for the robot’s services; in this study, the primary function demonstrated was a prompt-based walkthrough of hand-washing and tea-making. Caregivers, on the other hand, were enthusiastic about the robot, and some showed interest in immediate acquisition of a robot of their own to ease their workload [48]. More human-like, expressive robots appear to elicit more favorable responses from patients. Another study of similar nature employed Brian 2.1, a robot centered around social and cognitive interaction, and observed the reactions of 46 elderly individuals. Researchers found that the participants liked the human-like voice of the robot most of all, and the companionship it provided by its presence, facial expressions and voice tone differences, as well as its humanoid appearance [49]. Another study involving Brian 2.1 deployed the design to a long-term care facility, with study participants including those with no cognitive impairment, as well as MCI and mild AD patients. The robot was most liked by participants for its lifelike emotional expression, conveyed through its facial expressions and voice modulation, as well as its other human-like behaviors. The subjects believed that the use of the robot was a good idea, and that they would like to use it again (4.71/5 and 4.53/5, respectively). Participants also had a more moderate, but still positive, view towards its utility (3.24/5–3.53/5) [45].

A study of a small focus group of elderly individuals living with mild cognitive impairment helped to summarize some of the concerns that would need to be addressed for wide-scale deployment of a robotic dementia-care solution. Namely, potential users’ preconceptions about robots, that they are bulky, emotionless hunks of metal, can make them opposed to a mechanical companion [50]. Indeed, greater familiarity with robots cultivates a more favorable opinion on how useful, intelligent, and sociable they can be among elderly individuals [51]. Furthermore, there is a tendency to believe that they are only useful for the severely impaired, with a sort of stigma attached to the idea of needing robot-facilitated care [43,50]. Perhaps introducing robotic companions to such individuals would help to alleviate some of these fears and garner a more receptive response. Regardless, it is clear that giving the assistive robot a lifelike “personality”, and the means to express that, are vital to patient acceptance of a robot companion. Positive social behaviors, such as the ability to convey emotions such as pleasure and friendliness, and personalization of a SAR to fit user preferences is also likely to be vital to getting the widest possible audience to accept their use [43].

### 2.4. Emotion Recognition and Response

It is important that a SAR is able to recognize emotions and respond accordingly. A robot that can gauge human emotions is able to detect sadness or physical pain and alert caregivers or authorities in case something seems wrong. It would also be able to monitor the progression of the disease, as the emotional response of AD patients seems to decline as the disease progresses [52]. Furthermore, it has been shown that dementia patients respond positively to socio-emotional interaction with a robot [42].

Even though emotion recognition seems a hard problem on the surface, there have been a number of advances in the field of computer vision that can render this task easy to implement. Especially with the recent increase of the use of Convolutional Neural Networks (CNN), the task of image recognition has become relatively trivial [53] and the accuracy of correctly recognizing images keeps increasing with new architectures [54]. Image recognition has a significant effect on being able to recognize emotions through a video feed of the patients. Researchers have been able to train such neural networks to recognize human emotions with high accuracy to facilitate human–computer interactions [55] and implement them in real-time [56]. Furthermore, a number of companies, such as Microsoft [57] and Affectiva [58] have publicly available Application Programming Interfaces (API) that easily enable emotion recognition for any application.

## 3. SAR Design—Overall Considerations

In designing companion robots for elderly care recipients, there is often a considerable disconnect between the two groups in terms of how they feel a robot should look and what features it requires [59]. Given that such features are important to patient acceptance and usage of the robotic companion, adopting a user-centric design process seems most prudent to the development of a successful SAR. However, given the deviations between what dementia patients believe they need, and what their caretakers think would be helpful [42], we have opted to include both caretakers and their patients under the umbrella of the end user for this preliminary design.

Such a SAR would primarily be targeted towards those with MCI up to those with moderate, but not severe, dementia. As noted, SARs can be detrimental to some of those with severe dementia [37], and though the utility of an SAR for severe dementia may be limited, there is considerable demand for assistance caring for those even at more mild stages of dementia, particularly among informal caregivers. As informal caregiving is expected to become more prevalent as Dementia cases rise and the number of available caregivers falls, alleviating some of the burdens they face is essential [60]. Though more severely impaired individuals may still benefit from a SAR, research on non-animal SARs featuring complex interactions is largely limited to those with more moderate cognitive disability. The robot is also targeted towards those who live in their own homes; the robot will feature a high degree of personalization that is optimal for a small number of users.

Though many of the more utilitarian functions desired in an assistive robot can be accomplished through current smart-home and other technologies, they fall short when it comes to creating a social companion. Having a physical body is integral to the development of trust and attachment to the robot [61,62], which is necessary especially in the face of user skepticism. According to the studies referenced above, patients and caretakers would ideally prefer a robot with the following characteristics: Humanoid appearance, chest height, ability to detect dangerous situations, such as gas leaks, turned-on stoves, and falls, and to alert relatives and emergency services in these scenarios. It would also be able to operate gas and electric appliances, as well as set reminders, and provide cognitive and social stimuli and socio-emotional interaction. Some subjects also mentioned the ability to pick up objects and patients.

In designing a SAR, we must first decide on an appropriate budget for the design. By 2030, the number of dementia patients is expected to reach 75 million, and the global cost of care is expected to be upwards of 2.5 trillion dollars [11,12]. The global average cost per patient was estimated to be around $17,483 per year in 2015 [63]. Conservatively, assuming that the lifespan of a robot is about 8 years, we can see that a price point in the range of $10,000 for a robot unit is under 10% of the total cost of care for the patient over the lifespan of the robot. When taking into account the potential for the robot to drastically cut down on human work hours and improve caretaking, while reducing the patients’ risk, this cost can be well-justified.

### 3.1. SAR Design—Mechanical Components

Full-size anthropomorphic robots not only have very large manufacturing costs (the typical cost is upwards of $10,000 USD for a >0.5 meter robot with a humanoid chassis [64]), but also present significant challenges with mapping and localization of unknown spaces and present risk of falling or dropping things on patients, all of which we want to avoid. In order to keep the manufacturing cost in the desired range and reduce the complexity, our initial design does not feature locomotion. Actuators comprise much of the cost of manufacturing a humanoid robot [65], and a non-motile design removes them from the equation. Our design of the SAR consists of a humanoid torso and a head. The torso houses most of the electronic components, such as the main computing unit, power supply unit, some sensors and speakers, and is designed to be seated on a tabletop. As seen in Figure 3, the height of the torso is 50 cm and the head around 20 cm, for a total height of 70 cm. On the surface of a regular table 80 cm high, this design would have to reach the chest-height size desired by most users [42]. The internal structure (main support frame) would be constructed of steel or a similar material to give structural rigidity to the robot and support the internal components. On the support frame would be attached the shelves for the electronic components, the torso panels which would be made by thermoformed plastic, as well as the head support. The head support would be attached to a servo or stepper motor to allow rotation of the head which will be made from plastic.

### 3.2. SAR Design—Electronics

For the face of the robot, we plan to implement an LCD touch screen which will allow for high-quality facial expressions of the robot, while lowering manufacturing costs, as the construction of a robotic face with the desired expressive ability is a complicated, expensive process [34]. Furthermore, it allows for the easy customization of the robot’s facial appearance, so that it can match the user’s cultural and other preferences. The screen can also be used to show useful information, reminders, or allow face-to-face communication with relatives and friends, as well as accept touch-based input for certain functions.

The SAR would also need a large assortment of sensors to monitor the patient. In particular, the head would house a camera, infra-red depth sensors, and a microphone, while the torso would house air-quality sensors, such as a combustible gas detector, a carbon monoxide and dioxide detector, as well as smoke and temperature sensors. The camera, microphone, and the infra-red sensors can not only help with communication with friends and relatives, but can also track the patient and register falls or distress. The rest of the sensors can trigger warnings and alerts whenever dangerous situations arise.

Since our SAR design is not able to move, but a highly needed feature is the control of electronic devices, we plan on implementing smart-home protocol communication so that the robot can control devices such as lights, heaters, the stove, or speakers through the internet, without needing to move. The socially assistive robot would be always connected to the internet and would be able to communicate with as large a variety as possible of existing smart home interfaces, in order to eliminate the need for additional hardware. We plan on furthermore enhancing interoperability with existing technologies through communication with health-monitoring devices, such as wearable technology and mobile phones in order to collect health data and monitor for any unusual or dangerous signals. Including compatibility with these devices allows for users to choose what they want to use, while combining their functions to a single, relatively easy to use interface. As ease of use is a considerable barrier to the use of smart-home technology in the elderly, where having the robot serve as a unified, voice-enabled interface may allow less technologically familiar users to take advantage of them [66].

The array of external sensors and apparatuses compatible with the robot will be optional, as some users may have concerns over privacy. Though the elderly persons surveyed note that it is beneficial to have smart-home technology for detecting falls and other dangers, they also express concerns over the presence of cameras in their homes. Notably, cameras with limited imaging capability (i.e., they can detect shadows or movements, but not features of the individual) are viewed more favorably, and can provide an alternative for those concerned about their privacy [66]. Other forms of health- and security-related monitoring are generally well-received amongst older persons if they conceivably result in greater security and hastened emergency-response times [67]. While monitoring is certainly useful for both dementia patents and their caregivers, the social aspect of a SAR is still of considerable importance, and should be available even if users do not want the extra sensors and smart-home utilities.

The computer controlling the SAR will be a raspberry pi or equivalent running a free, open-source Linux Ubuntu and Robot Operating System (ROS) on top of which will be overlaid the additional software modules. This combination should be able to perform all the necessary functions of our robot while being very economically efficient, especially during the development and prototyping phases.

In order to provide power, a small power supply will be included in the torso, which will be connected directly to a power outlet. The power needs of the robot should always be under 150W, so a 200W supply will be more than adequate. Since the robot will not be moving, no battery is needed.

### 3.3. SAR Design—Software and Features

One of the key components of the robot is interaction with patients. The communication is designed to be primarily voice-based via the speakers and the microphones. The robot will accept voice commands through voice recognition and respond through the speakers. The ability to answer user questions and have non-repetitive dialogue with them is key to the creation of a conversational partner that users can bond with [68,69], and a robot that is more engaged itself will in turn encourage the user to engage [70]. The NLP module itself is something that will need to be developed with the userbase, elderly dementia patients, in mind, as the users are more error-prone than the general population and are liable to become confused due to misinterpreted speech. Furthermore, mistakes and response-latency can greatly impair the flow of communication. Machine learning and other data analytics techniques are essential for tuning speech behavior to the user’s behavior, minimizing latency and the frequency of errors [71]. The screen will also be an important component in communication. Not only will it respond to the user through facial expressions and presenting relevant information, but it will also accept touch-input to perform functions, such as calling friends and relatives, or changing settings.

Another important feature is the robot’s ability to recognize the emotions of the patient, through artificial emotional intelligence. The robot can then tailor its responses according to the user’s feelings and emotional state. This has recently become possible through advances in deep learning and neural networks that can decode a person’s feelings from a simple video feed [55,56]. The feature can be easily implemented through the use of AI emotion recognition modules, such as Microsoft’s, or Affectiva’s image-based emotion recognition APIs. Artificial Intelligence will also be used for object recognition, face recognition, and tracking. Using deep learning, the task of identifying people can be easily performed and can significantly improve the robot’s security features. For example, the SAR will be able to identify if an unknown person has entered the house, or if a person is the patient’s friend.

Through the settings and Machine Learning (ML) technologies, we aim to give the robot a high degree of personalization. Through settings such as face characteristics (skin tone, face shape, facial hair) as well as voice customization (male, female accents), the robot will be very personalized from the first use. Personalization should be possible through an interface if a more capable user or a caretaker wishes, or as the final stage of completing the robot for use by the manufacturer, as was done for the “Ryan” robot, based on the requests of the recipient [34]. After that, the robot will be able to tweak its responses, tone, and facial expressions over time depending on what the feedback from the patient is. Using machine learning, it is easy for the robot to learn what type of response makes the patient feel more at ease and how to better fit the needs of the patient.

Furthermore, through the software design, the robot can play games with the patients in order to provide cognitive and social stimuli. By tracking the responses of the patient to everyday questions, their ability to play games, as well as their emotional state, such a robot would also be very valuable in tracking the progression of the disease. Tying these games to the robot itself may promote their use if the user enjoys the presence of the robot.

Personalization will be a key feature of the robot. The robot will take initiative and try to stimulate conversation or encourage the patient to partake in the cognitive games, though the extent to which the robot tries to initiate or remind the user about these games/tasks will depend on user preferences. However, for both activity reminders and conversation initiation, there is the potential for a user to perceive the interaction as annoying. Ideally, the robot will be able to recognize signs that its input is unwanted and tune its behavior based on user responses. Manual settings could limit the number of conversations initiated by the robot, or set them to only occur at certain times.

## 4. Conclusions

The global trend of increasing life expectancy and decreasing fertility rates means that in the near future we will need more medical staff to take care of the large older population which suffers from AD and ADRD. There is a huge economic and humanitarian incentive to explore the use of socially assistive robots in the care of older individuals suffering from dementia. Available research provides encouraging results for the use of SARs and the patients’ response to them, as well as key functions that the robots would ideally feature. Current solutions do not provide the features that are most requested by AD and dementia patients and caregivers, even though technology is at a level that enables implementation of almost all these features that are important to people dealing with AD and ADRD patients on a daily basis.

Through our explanation of the design concept, it is shown that a multi-purpose SAR that performs almost all the functions desired by AD and ADRD patients and their caregivers is indeed possible. The initial design outlined here demonstrates that socially assistive robots should help to greatly reduce the cost, as well as increase the quality of care for AD and ADRD patients, by providing assistance and companionship to patients that is infeasible to provide. Such robots are very realistic and can be easily prototyped and manufactured at very affordable prices. The methods used for our design process are entirely user-centered, based on surveys and studies performed on real patients and caretakers. In the future steps of our design, we plan on interacting directly with patients and caregivers to make sure that the robotic solution can be as helpful as possible to patients and their families. Though SARs may not be viable for all persons with dementia, any significant reduction of the burden on caregivers would be a great boon that would potentially free up resources for more severe AD/dementia cases.

## Figures and Tables

**Figure 1 healthcare-08-00073-f001:**
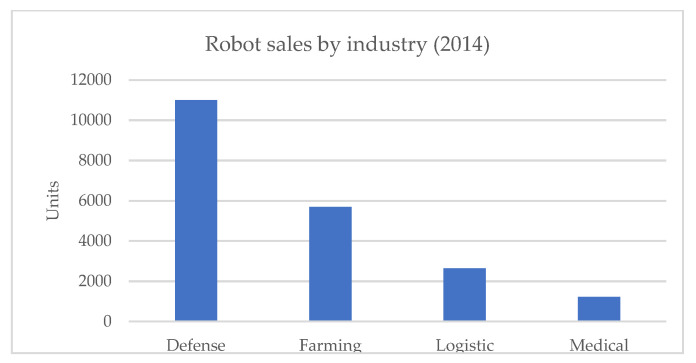
Service robots per industry, for the year 2014. data was obtained from the executive summary of world robotics 2015 [4].

**Figure 2 healthcare-08-00073-f002:**
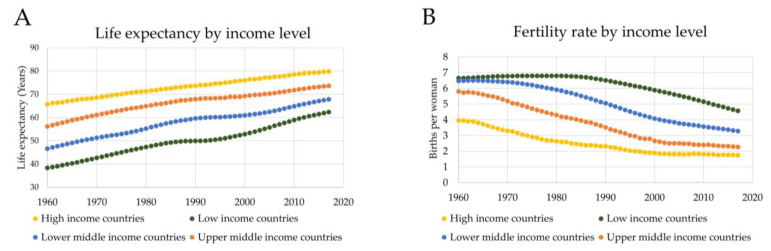
Life expectancy and fertility rates from 1960-2017 based on country income level. (A) Life expectancy per year by country income level [16]; (B) birth rates per year by country income level [17].

**Figure 3 healthcare-08-00073-f003:**
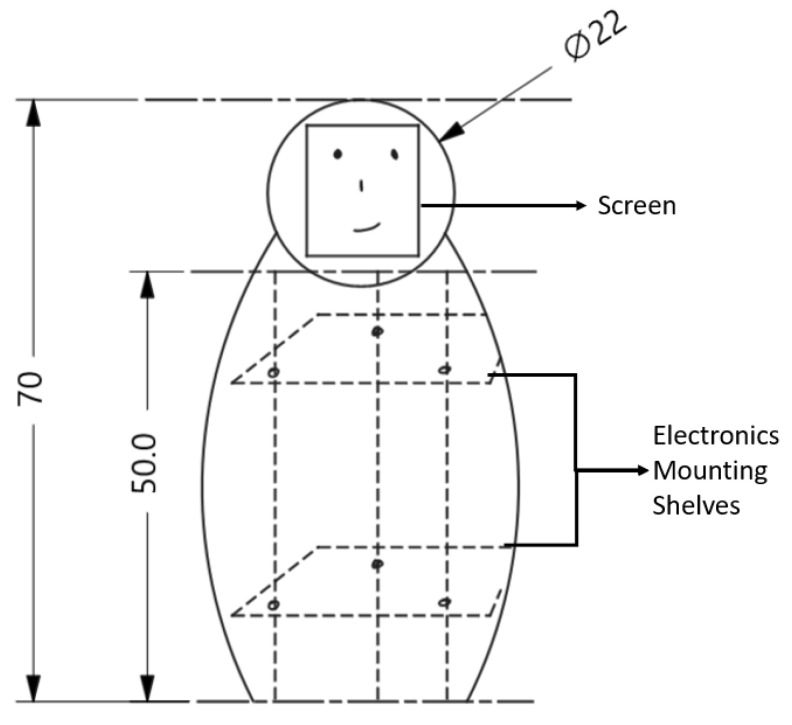
Socially Assistive Robot (SAR) design sketch. The total height will be ~70 cm, so that it is around chest height when placed on a table. This design is non-motile, with an LED-screen face on the robot’s head that doubles as a touch-screen for certain interactions. Electronics will be housed in the torso region ~50 cm tall.

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
