# Peer review of "Designing Socially Assistive Robots for Alzheimer’s Disease and Related Dementia Patients and Their Caregivers: Where We Are and Where We Are Headed"

_healthcare, 2020, doi:10.3390/healthcare8020073_

Round 1

Reviewer 1 Report

This paper presents a proposal for the construction of a social robot focused on interacting with elder people. This is an important topic considering that the world’s population is aging in the last years. However, there are certain aspects in which the authors could still provide further improvement.

Firstly, there are not clear the current contributions of the robot in comparison with other existing robots. To clarify this issue, it would be interesting to carry out an analysis of the different functionalities that the authors want to cover with the robot, differentiating them from the possibilities of a home automation system, and the different challenges that may arise, as well as specifying the type of interaction and where it will be carried out (care centers or home, and in the second case, which room). In the same way, it would be interesting to clarify how the personalization of the robot to the user is going to be done and if it is going to be necessary for the development of some kind of interface to be able to record the user's data.

On the other hand, to explain the current state of the art, it would be important to analyze in more detail the current robots oriented to elders or that have been used to interact with them. These robots must cover different functionalities, not only from the affective point of view and based on non-verbal communication, like Aibo, NeCoRo or Paro. In the case of existing studies, details should be given on whether participants had any knowledge of the possibilities of a robot, and what type of functionalities were covered.

Finally, it would be interesting to include the collaboration of at least caregivers for the design of functionalities and the robot itself, even before the construction of the robot.

Author Response

Reviewer 1’s comments and responses

This paper presents a proposal for the construction of a social robot focused on interacting with elder people. This is an important topic considering that the world’s population is aging in the last years. However, there are certain aspects in which the authors could still provide further improvement.

Firstly, there are not clear the current contributions of the robot in comparison with other existing robots. To clarify this issue, it would be interesting to carry out an analysis of the different functionalities that the authors want to cover with the robot, differentiating them from the possibilities of a home automation system, and the different challenges that may arise, as well as specifying the type of interaction and where it will be carried out (care centers or home, and in the second case, which room). In the same way, it would be interesting to clarify how the personalization of the robot to the user is going to be done and if it is going to be necessary for the development of some kind of interface to be able to record the user's data.

Response: For this preliminary design, we do not have the capacity to analyze the different functions individually; furthermore, the studies we discuss in this manuscript also discuss their respective robots largely holistically, rather than breaking it down by feature. Though some functions could indeed be carried out by home-automation systems, the social aspects are greatly augmented by the presence of a physical body, which we have mentioned in the text (line 272). All personalized aspects will be modifiable via an interface with the robot itself, that way more cognitively capable individuals or a caretaker can alter anything they wish. However, particularly for initial testing, personalization will be pre-set based on user requests. This detail has been added to the text (line 351).

On the other hand, to explain the current state of the art, it would be important to analyze in more detail the current robots oriented to elders or that have been used to interact with them. These robots must cover different functionalities, not only from the affective point of view and based on non-verbal communication, like Aibo, NeCoRo or Paro. In the case of existing studies, details should be given on whether participants had any knowledge of the possibilities of a robot, and what type of functionalities were covered.

Response: We have gone into greater detail on the features of the robots used in the studies we reference. Unfortunately, most of the studies do not go into detail about what participants were told beforehand, regarding the possibilities of robots, other than instructions regarding their own robot. however, we have mentioned how preconceptions about robots can discourage their acceptance, while experience with them makes one more open to engaging with it.

Finally, it would be interesting to include the collaboration of at least caregivers for the design of functionalities and the robot itself, even before the construction of the robot.

Response: While we had emphasized our intent of a user-oriented design process, we included caregivers under the “user” umbrella but failed to mention this in the text (this has since been changed (lines 39 & 333)). As noted in some of the studies we mention, the behaviors and features requested by elderly care-recipients and the caretakers are not entirely aligned, and both parties have valuable input.

Reviewer 2 Report

Dear Authors,

Thank you very much for this very interesting paper. Your article is generally well-written, and would contribute to the debates and discussions concerning the development and use of robots in health and social care, if the following amendments are successfully made. 

Here are questions and requests for changes (including minor errors).

First of all, the title of this article is somewhat misleading, especially as part of this article sketches out your original design of an SAR (but without any reference to existing literature/R&D).

The gap between the actual review section (up to Section 3) and Section 3 should be filled by adding more information which includes the basis upon which this design was carried out. 3.1. does not have any references either. 

In addition, what about an alternative method such as a combination of existing tools and devices (for example, a robot connected to a monitoring sensor)? Why does it have to be a stand-alone product? 

The title ("Socially Assistive Robots for Alzheimer’s Disease and Related Dementia Patients and Their Caregivers: Where we are and where we are headed") should be revised to reflect what this article is about. For example, add a word 'Designing' before the current one.

Second, how did the authors identify the literature/existing research? A few sentences should be added, explaining how they did it (e.g. the inclusion/exclusion criteria)  

p.1, l. 19 -  "Herein, we propose a sketchy design for such SARs."

It should state how this was done (based on existing literature and its findings, for example).

p.2, l.33/34 - this sentence does not seem complete.

Furthermore, the introductory section should explain more clearly how this article is structured. In particular, why and how the themes for Section 2 (2.1-2.4) emerged/were selected? 

Currently, there is no mention of Figure 2 in the main text. Figure 2 should be mentioned where appropriate, perhaps, somewhere on p.2.

p.3, l.67, [15]3 --- remove 3.

p.4, l.120 - remove RAA, as this acronym is not used elsewhere. 

Quite a number of papers, relevant to this topic, are not referenced.

Liang A, Piroth I, Robinson H. A pilot randomized trial of a companion robot for people with dementia living in the community.J Am Med Dir Assoc 2017;18: 871–878.7

de Graaf MMA, Allouch SB, Klamer T. Sharing a life with Harvey:exploring the acceptance of and relationship-building with a social robot. Comput Human Behav 2015;43:1–14.8

Moyle W, Jones CJ, Murfield JEet al. Use of a robotic seal as a therapeutic tool to improve dementia symptoms: a cluster-randomized con-trolled trial.J Am Med Dir Assoc 2017;18: 766–773.

Obayashi K, Kodate N,Masuyama S. Measuring the impact of age, gender anddementia on communication-robot interventions in residential care homes. Geriatr. Gerontol. Int. 2020;1–6.

Pilotto A, D’Onofrio G, Benelli Eet al. Information and communication technology systems to improve quality of life and safety of Alzheimer’s disease patients: a multicenter international survey. J Alzheimers Dis 2011;23: 131–141.

If the paper is focused on the user-centred design, it should state this at a much earlier stage in the article and divide the paper more clearly along those lines (first, what we already know regarding the effects, etc., and second, the design components/process).

If the above-mentioned issues are addressed and rectified, I believe that this will be publishable.

Sincerely,

Reviewer

Author Response

Reviewer 2’s comments and responses

Dear Authors,

Thank you very much for this very interesting paper. Your article is generally well-written, and would contribute to the debates and discussions concerning the development and use of robots in health and social care, if the following amendments are successfully made. 

Here are questions and requests for changes (including minor errors).

First of all, the title of this article is somewhat misleading, especially as part of this article sketches out your original design of an SAR (but without any reference to existing literature/R&D).

Response: We have revised the title in accordance with your suggestion.

The gap between the actual review section (up to Section 3) and Section 3 should be filled by adding more information which includes the basis upon which this design was carried out. 3.1. does not have any references either.

Response: We have added a brief introduction to the general design process to ease the transition between the review section and the design section (line 327)

In addition, what about an alternative method such as a combination of existing tools and devices (for example, a robot connected to a monitoring sensor)? Why does it have to be a stand-alone product?

Response: The benefit of a stand-alone product mainly pertains to the social aspect. While many of the utilitarian (e.g. monitoring, reminders, etc.) features can be accomplished through other technologies, the social aspects are more nuanced. The development of trust and bond-formation with the robot is benefited by a physical presence (i.e. a body). We have added a (referenced) note of this in the text (line 279).

“The title ("Socially Assistive Robots for Alzheimer’s Disease and Related Dementia Patients and Their Caregivers: Where we are and where we are headed") should be revised to reflect what this article is about. For example, add a word 'Designing' before the current one.”

Response: We have revised the title according to your suggestion, adding the word “Designing” to the beginning. Thank you for the suggestion.

Second, how did the authors identify the literature/existing research? A few sentences should be added, explaining how they did it (e.g. the inclusion/exclusion criteria) 

Response: We have added a brief explanation of the general search-strategy to the introduction (line 43).

“p.1, l. 19 -  "Herein, we propose a sketchy design for such SARs."

It should state how this was done (based on existing literature and its findings, for example).”

Response: We have revised the abstract as requested, and have added a statement on the premise and basic methodology of our design (line 19).

“p.2, l.33/34 - this sentence does not seem complete.”

Response: We have corrected the sentence, thank you for pointing it out (line 35).

Furthermore, the introductory section should explain more clearly how this article is structured. In particular, why and how the themes for Section 2 (2.1-2.4) emerged/were selected?

Response: We have added more detail about the intent and structure of the manuscript to the introduction of the paper.

“Currently, there is no mention of Figure 2 in the main text. Figure 2 should be mentioned where appropriate, perhaps, somewhere on p.2.”

Response: We have added a mention of Figure 2 in reference to increases in life expectancy and declining birthrates (line 50).

“p.3, l.67, [15]3 --- remove 3.

p.4, l.120 - remove RAA, as this acronym is not used elsewhere.”

Response: We have corrected these errors, thank you for pointing them out.

Quite a number of papers, relevant to this topic, are not referenced.

Liang A, Piroth I, Robinson H. A pilot randomized trial of a companion robot for people with dementia living in the community.J Am Med Dir Assoc 2017;18: 871–878.7

de Graaf MMA, Allouch SB, Klamer T. Sharing a life with Harvey:exploring the acceptance of and relationship-building with a social robot. Comput Human Behav 2015;43:1–14.8

Moyle W, Jones CJ, Murfield JEet al. Use of a robotic seal as a therapeutic tool to improve dementia symptoms: a cluster-randomized con-trolled trial.J Am Med Dir Assoc 2017;18: 766–773.

Obayashi K, Kodate N,Masuyama S. Measuring the impact of age, gender anddementia on communication-robot interventions in residential care homes. Geriatr. Gerontol. Int. 2020;1–6.

Pilotto A, D’Onofrio G, Benelli Eet al. Information and communication technology systems to improve quality of life and safety of Alzheimer’s disease patients: a multicenter international survey. J Alzheimers Dis 2011;23: 131–141.

Response: We have added these references where relevant, among others. Thank you for providing them.

If the paper is focused on the user-centred design, it should state this at a much earlier stage in the article and divide the paper more clearly along those lines (first, what we already know regarding the effects, etc., and second, the design components/process).

Response: We have expanded the introduction to include a section on the structure of the paper, and what the paper proposes.

If the above-mentioned issues are addressed and rectified, I believe that this will be publishable.

Sincerely,

Reviewer

Response: Thanks for the reviewer. We have tried our best to address your critiques.

Round 2

Reviewer 1 Report

In the previous review, the authors were notified to make improvements in different aspects. Although they indeed made certain changes, the main innovation of the robot introduced remains unclear.
As the paper simply presents the proposal of a robot designed to interact with elder people, without having carried any complementary study, it would be necessary to analyze at least certain aspects that have not been covered: what type of interaction will be carried out from the robot initiative point of view, or a list of robot functionalities, not from a general point of view (these functionalities will justify the robot design).
But perhaps the most important aspect to be defined is the target users. In the paper, the degree of cognitive impairment of the user is not clear. This is essential to understand the type of interaction that must be carried out. According to this aspect, the robots introduced in the state of the art should also be classified, considering that a robot for a mild cognitive impairment (providing a more complex interaction) has to differ with another for a moderate one (where non-verbal language prevails).
Finally, it seems that the authors present the design of the robot from a utopian viewpoint, in which the interaction between an inexperienced older person and a robot with limited capabilities can interact without problems. Taking this into consideration, it is still necessary to analyze certain concerns that may arise during the interaction of the robot, to be taken into account in its design: inconsistency in the recognition of user emotions, and therefore the behavior of the robot according to recognition, lack of user attention, how many home automatic sensors would be necessary and if the elderly person could feel uncomfortable about it, etc.

Author Response

Review 1’s critiques and authors' responses:

“In the previous review, the authors were notified to make improvements in different aspects. Although they indeed made certain changes, the main innovation of the robot introduced remains unclear.
As the paper simply presents the proposal of a robot designed to interact with elder people, without having carried any complementary study, it would be necessary to analyze at least certain aspects that have not been covered: what type of interaction will be carried out from the robot initiative point of view, or a list of robot functionalities, not from a general point of view (these functionalities will justify the robot design).”

Response: We have added a section on robot initiative and the personalization of the robot-user initiation of activity (line 459). As noted, this is a preliminary, so features are not fully elaborated; however, we have gone into more detail on some of the social and personalization aspects that are central to the design.

“But perhaps the most important aspect to be defined is the target users. In the paper, the degree of cognitive impairment of the user is not clear. This is essential to understand the type of interaction that must be carried out. According to this aspect, the robots introduced in the state of the art should also be classified, considering that a robot for a mild cognitive impairment (providing a more complex interaction) has to differ with another for a moderate one (where non-verbal language prevails).”

Response: We have added more information on the cognitive state of the participants in the studies we reference, particularly in regard to the humanoid SARs. We have also noted that our design targets those with up-to moderate dementia, particularly those still able to live at home.

“Finally, it seems that the authors present the design of the robot from a utopian viewpoint, in which the interaction between an inexperienced older person and a robot with limited capabilities can interact without problems. Taking this into consideration, it is still necessary to analyze certain concerns that may arise during the interaction of the robot, to be taken into account in its design: inconsistency in the recognition of user emotions, and therefore the behavior of the robot according to recognition, lack of user attention, how many home automatic sensors would be necessary and if the elderly person could feel uncomfortable about it, etc.”

Response: We have added sections on potential privacy concerns (line 368), as well as how inconsistent communication can be handled (line 388). We have also noted that the technology may not be suitable for all individuals with dementia in the conclusion.

Reviewer 2 Report

Dear Authors,

Thank you very much for your thorough treatment of my comments.

Happy to sign this off.

Reviewer

Author Response

Review 2’s critiques and authors’ responses:

“Dear Authors,

Thank you very much for your thorough treatment of my comments.

Happy to sign this off.

Reviewer”

Response: We thank this reviewer’s encouragement and approval of our manuscript.